# Antiviral Effect and Mechanism of Edaravone against Grouper Iridovirus Infection

**DOI:** 10.3390/v15112237

**Published:** 2023-11-10

**Authors:** Jihui Kuang, Mingzhu Liu, Qing Yu, Yuan Cheng, Jing Huang, Shuyu Han, Jingu Shi, Lin Huang, Pengfei Li

**Affiliations:** 1School of Resources, Environment and Materials, Guangxi University, Nanning 537100, China; jihkaung@163.com; 2Guangxi Key Laboratory of Aquatic Biotechnology and Modern Ecological Aquaculture, Guangxi Engineering Research Center for Fishery Major Diseases Control and Efficient Healthy Breeding Industrial Technology (GERCFT), Guangxi Academy of Marine Sciences, Guangxi Academy of Sciences, Nanning 530022, China; liu408149595@163.com (M.L.); yu_qing1990@163.com (Q.Y.); chengyuan19872022@163.com (Y.C.); huangqinglin1220@163.com (J.H.); henry0779@163.com (S.H.); sjg77@163.com (J.S.); 3China-ASEAN Modern Fishery Industry Technology Transfer Demonstration Center, Beibu Gulf Marine Industrial Research Institute, Guangxi Academy of Marine Sciences, Nanning 530022, China; 4Guangxi Fisheries Technology Extension Station, Nanning 530022, China

**Keywords:** grouper iridovirus, edaravone, antiviral effect

## Abstract

Singapore grouper iridovirus (SGIV) is a virus with high fatality rate in the grouper culture industry. The outbreak of SGIV is often accompanied by a large number of grouper deaths, which has a great impact on the economy. Therefore, it is of great significance to find effective drugs against SGIV. It has been reported that edaravone is a broad-spectrum antiviral drug, most widely used clinically in recent years, but no report has been found exploring the effect of edaravone on SGIV infections. In this study, we evaluated the antiviral effect of edaravone against SGIV, and the anti-SGIV mechanism of edaravone was also explored. It was found that the safe concentration of edaravone on grouper spleen (GS) cells was 50 µg/mL, and it possessed antiviral activity against SGIV infection in a dose-dependent manner. Furthermore, edaravone could significantly disrupt SGIV particles and interference with SGIV binding to host cells, as well as SGIV replication in host cells. However, edaravone was not effective during the SGIV invasion into host cells. This study was the first time that it was determined that edaravone could exert antiviral effects in response to SGIV infection by directly interfering with the processes of SGIV infecting cells, aiming to provide a theoretical basis for the control of grouper virus disease.

## 1. Introduction

Grouper (*Epinephelus* spp.) is an important cultured species in the aquaculture industry. The China Fisheries Statistical Yearbook [1] shows that the production of grouper culture in China reached 192,045 tons in 2020. With the increasing scale of grouper culture, aquatic environmental problems have become more and more serious, and the deterioration of the environment has led to an increasing probability of pathogen outbreaks, which now seriously threatens the sustainable development of the grouper culture industry [2].

Iridescent virus is one of the most damaging pathogenic viruses in aquatic diseases and has been isolated from more than 100 fish species worldwide. Singapore grouper iridovirus (SGIV) belongs to the iridovirus family [3]. Infection of grouper with SGIV results in some typical clinical signs, such as spleen and liver enlargement and mass mortality, within 1–2 weeks [4]. At present, vaccines and drugs are some of the effective means to prevent and control SGIV [5]. It is shown that two previous vaccines for SGIV offer some protection against SGIV infection in grouper [6]. Liu et al. found that a subunit vaccine containing the recombinant major capsid protein (*MCP*) of Taiwan grouper iridovirus (TGIV) was effective in protecting grouper, with a relative survival rate of 86% [7]. However, the protective effect of vaccines must be carried out before infection in order to produce effective immune effect. Thus, vaccines only can be used as a means for virus prevention. On the other hand, problems of inactivation, strict requirements for storage conditions, and unclear doses seriously affect the application of the vaccine in the aquaculture industry [7,8]. Therefore, we need to develop antiviral agents that are safe for aquaculture and simple to handle.

It has been reported that chemical drugs are still the most powerful “weapon” in response to virus infections, due to their single component and stable structure [9]. Many chemical drugs have the ability to restrict virus infection, such as ribavirin, which has an inhibitory effect on human norovirus [10]; it also can inhibit the replication of HSV-1, HSV-2, culex virus, rotavirus, etc. [11]. Moreover, the receiver can also limit the replication of the ebola virus [12]. Brouwers et al. found that chloroquine inhibited HIV replication in a dose-dependent manner in vitro [13]. Additionally, antiviral chemical drugs against aquatic animal viruses were increasingly studied. For example, ribavirin inhibits largemouth black bass elasmobranch virus (MSRV) in vitro [14], and it also inhibits viral hemorrhagic septicemia virus (VHSV) [15] and tilapia lake virus (TiLV) [16]. Metformin can interfere with SGIV binding to host cells and replication, as well as impairing SGIV particles [4]. The above reports indicate that the use of drugs to treat viral infections has great potential for application.

Edaravone, a member of the chemical class, is derived from pyrazolone and was first marketed in Japan in 2001 for the treatment of cerebral infarction [17,18]. It has been confirmed that edaravone has a wide range of applications, including the treatment of neurological disorders, central nervous system diseases (Alzheimer’s disease, ALS), cardiovascular dysfunction, chronic obstructive pulmonary disease (COPD), and drug-induced chronic kidney injury [19]. Edaravone shows high and balanced antioxidant activity compared with other antioxidants such as uric acid, glutathione and Trolox, and has a broad applicability to a wide range of free radicals [20]. Moreover, the effects of anti-inflammatory, anti-cytokine, immunomodulatory, anti-apoptotic, anti-necrotic, anti-fibrotic, membrane stabilization, protection against lung surface active substances and protection against I/R-induced injury in several organs are also the characteristic of edaravone [21]. The report showed that edaravone inhibits the toxicity of cationic liposomes to cells by eliminating the cell-damaging ROS generated during lipofection [22]. Edaravone increases zonula occludens-1 protein (ZO-1) expression at the mRNA and protein levels. Simultaneously, edaravone inhibits permeability changes in human pulmonary microvascular endothelial cells (PMVEC) by enhancing vascular endothelial cadherin to intervene in the treatment of ARDS [23]. The use of edaravone greatly improved CSDS-induced depression and anxiety behaviors. In addition, edaravone significantly attenuated CSDS-induced neuronal loss, microglial activation, astrocyte dysfunction, oxidative stress damage, energy metabolism and activation of pro-inflammatory cytokines in the hippocampus (Hip) and mPFC of mice [24].

However, there are no reports on the use of edaravone for the control of viral diseases in grouper. Our object in this study was to investigate the role and the mechanism of the action of edaravone in SGIV infection in vitro, aiming to address the need for an effective therapeutic agent for iridovirus infection during aquaculture. 

## 2. Materials and Methods

### 2.1. Cells and Viruses

Grouper spleen (GS) cells were isolated and preserved in our laboratory. GS cells were cultured at 28 °C in Leibovitz’s-15 medium (Gibco, Grand Island, NY, USA) containing 10% fetal bovine serum (Thermo Fisher Scientific, Waltham, MA, USA) [25]. The SGIV was isolated from hybrid grouper reared in Guangxi (*Epinephelus fuscoguttatus* ♀ *× Ephinephelus lanceolatus* ♂). The aptamer LYGV1, which can specifically identify SGIV-infected GS cells [26], was screened using the SELEX technique. The aptamer LYGV1 was labeled with 6-carboxyfluorescein (FAM) and synthesized by Sangon Biotech (Shanghai, China). Edaravone was purchased from Chengdu Herbpurify Co., Ltd. (Chengdu, China).

### 2.2. Determination of the Safe Concentration of Edaravone

The cells (1 × 10^5^ cells/well) were seeded into 96-well plates and incubated in a cell culture incubator at 28 °C for 18 h. Cells were incubated with edaravone (50, 125, 250, 500, 1000 and 2000 µg/mL) or L-15 medium alone for 48 h at 28 °C. The cell morphology was observed using an inverted microscope (Nikon, Ts2, Japan, Tokyo Metropolis) for 48 h, and then 10 µL/well of CCK-8 solution was added to the cells and incubated at 28 °C for 4 h. The effect of edaravone on the cell viability was determined by measuring the absorbance at 450 nm using an enzyme-labeled instrument (Thermo, Waltham, MA, USA). The cell viability was calculated according to the following formula: Cell viability rate = Absorbance of cells in treatment group/absorbance of cells in control group × 100%.

To further determine that 50 μg/mL is the safe cell concentration of edaravone, we tested the cytotoxicity of edaravone using laser scanning confocal microscopy (LSCM, Nikon, C2, Japan, Tokyo Metropolis) observation. Briefly, the cells (1 × 10^5^ cells/well) were seeded in a 35 mm glass-bottom dish and incubated in a cell incubator at 28 °C for 18 h. Edaravone, with a concentration of 50 μg/mL, was added to incubate the cells for 48 h; the cells incubated with L15 medium were used as control. After that, the cells were fixed with 4% paraformaldehyde for 1 h at 4 °C, and washed three times, and subsequently incubated with keratin (Anti-pan Cytokeratin). Finally, keratin was labeled with FAM-FITC. The cell morphology was observed using laser scanning confocal microscopy (LSCM, Nikon, C2, Japan, Tokyo Metropolis).

### 2.3. Confirmation of Gene Expression by RT-qPCR

The cells (8 × 10^5^ cells/well) were seeded into 12-well plates and incubated in a cell culture incubator at 28 °C for 18 h. The cells were treated with SGIV or SGIV+edaravone (50, 25, 12.5 µg/mL) for 48 h. The cytopathic effect of each group was observed using a light microscope, and the cells from each group were collected for total RNA extraction to detect the expression of the major capsid protein (*MCP*) and vesicle protein (*VP19*) of SGIV by RT-qPCR, the β-actin gene (*β-Actin*) was used as an internal reference [27]. The primers used for RT-qPCR are listed in Table 1.

### 2.4. Aptamer (LYGV1)-Based Fluorescent Molecular Probe (AFMP) Monitoring of SGIV Infection

To further confirm the antiviral efficacy of edaravone, we used the FAM fluorescently labeled aptamer LYGV1 to detect SGIV infection using flow cytometry. In short, the cells were treated with SGIV or SGIV+edaravone (50, 25, 12.5 µg/mL), and cells treated with L15 medium alone were used as control; 48 h afterward, the cells in each group were collected to be incubated with FAM-labeled aptamer LYGV1 for 30 min at 4 °C. Then they were washed twice with PBS. Finally, the cells were resuspended and analyzed using flow cytometry. 

### 2.5. Antiviral Mechanism of Edaravone against SGIV Infection: Effect of Edaravone on SGIV Particles

The cells (8 × 10^5^ cells/well) were seeded into 12-well plates and incubated in a cell incubator at 28 °C for 18 h. Edaravone (50 µg/mL) and SGIV were co-incubated at 4 °C for 2 h. The virus was collected by centrifugation at 25,000 g for 30 min at 4 °C and resuspended in 100 µL TN buffer. Subsequently, 10 µL of the treated virus particles was added to the cells in the 12-well plate. SGIV without edaravone treatment was the control group. After 48 h, the cells of each group were collected to extract total RNA for the detection of SGIV *MCP* and *VP19* expression by RT-qPCR. Moreover, we also characterized SGIV infection by viral titer determination. The viral titers of SGIV in each group were determined by 50% tissue culture infectious dose (TCID_50_), as previously described. 

### 2.6. Antiviral Mechanism of Edaravone against SGIV Infection: Effect of Edaravone on SGIV Binding to Host Cells

The cells (8 × 10^5^ cells/well) were seeded into 12-well plates and incubated at 28 °C for 18 h. The cells were treated with edaravone (50 µg/mL) +SGIV for 30 min at 4 °C. After removing the supernatant, the cells were washed twice with L15 medium and then cultivation continued for 12 h at 28 °C. The cells of each group were collected to extract total RNA for the detection of SGIV *MCP* and *VP19* expression by RT-qPCR. The cells treated with SGIV only were used as control.

We also performed the assay using flow cytometry by treating edaravone (50 µg/mL) with Cy5-labeled SGIV (Cy5-SGIV) [27] for 2 h at 4 °C, while the control group was treated with Cy5-SGIV only. After 2 h, the cells were collected from each well and washed twice with PBS. Finally the results were analyzed using flow cytometry.

### 2.7. Antiviral Mechanism of Edaravone against SGIV Infection: Effect of Edaravone on SGIV Invading the Host Cells

The cells (8 × 10^5^ cells/well) were seeded into 12-well plates and incubated at 28 °C for 18 h. SGIV was added to the cells for 30 min adsorption at 4 °C. After that, the cells were washed twice with L15 medium and then treated with edaravone (50 µg/mL) for 2 h at 28 °C, and the cells treated with L15 medium were used as control. Subsequently, the cells were harvested from each group after 12 h incubation at 28 °C for total RNA extraction, and the expression of SGIV *MCP* and *VP19* was detected by RT-qPCR.

Furthermore, the effect of edaravone on SGIV invading the host cells also validated using flow cytometry analysis. The cells were treated with Cy5-SGIV at 4 °C for 2 h and washed twice with L15 medium. Following that, the cells were treated with edaravone (50 µg/mL) or L15 medium for 2 h at 28 °C. The cells were washed twice with PBS after digesting with proteinase K, and then subjected to flow cytometry for analysis.

### 2.8. Antiviral Mechanism of Edaravone against SGIV Infection: Effects of Edaravone on SGIV Replication in Host Cells

The cells (8 × 10^5^ cells/well) were seeded into 12-well plates and incubated at 28 °C for 18 h. The cells were treated with SGIV at 4 °C for 30 min and then incubated at 28 °C for 2 h to allow SGIV to enter the host cells. After washing twice with L15 medium, the cells were incubated with edaravone (50 µg/mL) for 10 h at 28 °C. The control group was SGIV-infected cells that were treated without edaravone. Subsequently, the cells were collected for total RNA extraction and the expression of SGIV *MCP* and *VP19* was examined by RT-qPCR.

### 2.9. Statistical Analysis

The data were presented as three independent experiments with mean ± SD. Results between groups were analyzed with one-way Student’s *t*-test in SPSS 16.0 software. Statistical significance was indicated by *, *p* < 0.05, and **, *p* < 0.01, respectively.

## 3. Results

### 3.1. Edaravone at a Safe Working Concentration Exhibited No Cytotoxic Effects

To investigate the safe working concentration of edaravone, edaravone at the concentration of 50, 125, 250, 500, 1000 and 2000 µg/mL was used to treat the GS cells. It was observed by microscopy that the cell morphology changed significantly at the concentrations higher than 50 µg/mL, while the GS cells grew well, with no change in morphology, at the concentration of 50 μg/mL (Figure 1A). Cell activity analysis also revealed that the concentration of 50 µg/mL had no effect on the cell activity, and the results were consistent with those of the light microscopy (Figure 1B).

The cytoskeleton is the main mechanical structure of cells and plays an important role in cell function. To further verify whether 50 μg/mL is the safe working concentration of edaravone, we made a thorough inquiry into the effect on the cytoskeleton of edaravone. We found that the cytoskeleton of the cells incubated with edaravone (50 μg/mL) remained normal compared with the control cells (Figure 1C), indicating that the safe concentration of edaravone did not cause any cytotoxic effects. These results were consistent with the observations from the previous CCK-8 assay, which confirmed that the safe working concentration of edaravone was ≤ 50 μg/mL

### 3.2. Inhibition Effects of Edaravone on SGIV Infection

To explore the effect of edaravone on SGIV infection, we treated SGIV-infected cells with edaravone. GS cells cultured with SGIV + edaravone (50, 25, 12.5 µg/mL) were the test group, and cells infected with SGIV alone were the control group. Cell morphology was observed using light microscopy. Few typical cytopathic effects (CPEs) were observed in cells cultured with SGIV + edaravone (50, 25 µg/mL) (Figure 2A). In addition, the expression of the SGIV *MCP* and *VP19* gene was significantly lower in the edaravone treated group (50 µg/mL, 25 µg/mL) compared to the control (Figure 2B). Similarly, flow cytometry assays showed a significant decrease in fluorescence intensity in cells treated with edaravone (Figure 2C).

### 3.3. Edaravone Possessed the Ability to Damage the SGIV Particles 

For investigating the antiviral mechanism of edaravone against SGIV infection, we firstly explored the destructive effect of edaravone on SGIV particles. The SGIV-infected cells treated with edaravone (50 µg/mL) were the test group, and those infected with SGIV only were the control group. It was shown that the viral *MCP* and *VP19* genes were significantly down-regulated in the test group compared to the control group (Figure 3A). Moreover, a significant decrease in the SGIV TCID_50_ in the test group was observed (Figure 3B), indicating that edaravone (50 µg/mL) has a destructive effect on SGIV particles.

### 3.4. Inhibitory Effects of Edaravone on SGIV Binding to Host Cells

To investigate whether edaravone has an effect on the binding of SGIV to host cells, the effect of edaravone on the binding of SGIV to host cells was first examined by RT-qPCR. The results revealed that the expression of SGIV *MCP* and *VP19* genes was significantly lower in the edaravone (50 µg/mL) treated group than that in the control group (Figure 4A). In addition, flow cytometry results showed that the Cy5 fluorescence signal was significantly reduced in cells treated with edaravone (50 µg/mL) compared with the control group (Figure 4B). These results suggested that edaravone (50 µg/mL) could interfere with the binding of SGIV to host cells.

### 3.5. Inhibitory Effects of Edaravone on SGIV Replication in Host Cells

The effect of edaravone on SGIV replication in host cells was analyzed by RT-qPCR. The results showed that the expression of SGIV *MCP* and *VP19* genes was significantly decreased in the cells treated with edaravone (50 µg/mL) compared to the control group (Figure 5), suggesting that edaravone could inhibit the replication of SGIV in the cells.

### 3.6. Inhibition Rate of Edaravone in Different Stages of SGIV Infection

The inhibition rate of edaravone on different stages of SGIV infection was examined by RT-qPCR. According to the expression of the viral *MCP* gene, it was shown that the inhibition rates of edaravone (50 µg/mL) on SGIV particles (Test 1), SGIV binding to host cells (Test 2), and SGIV replication (Test 3), were 38%, 41%, and 92%, respectively (Figure 6A). Similar results appeared in the expression of the SGIV *VP19* genes detection at different stages of SGIV infection (Figure 6B), revealing that edaravone (50 µg/mL) has good antiviral activity against SGIV.

## 4. Discussion

SGIV is a highly pathogenic iridovirus, which brings huge losses to the healthy cultivation of grouper. Currently, there are few drugs approved for the treatment of grouper infected with SGIV. Given that edaravone has many inexpensive and readily available formulations for clinical use, our goal in this study is to explore the effects of edaravone on SGIV infection. 

Edaravone is a pyrazolone derivative. Pyrazolinone scaffolds are found naturally in many alkaloids and various drugs [18], and their wide range of biological applications include anti-inflammatory [28,29], anticancer [30], analgesic [31], anti-diabetic [32], anti-microbial [33,34], antioxidant [35,36], active binding and enzyme inhibition [37]. Antipyrin, a derivative of pyrazolone, and its derivative Jodantipyrin have shown good antiviral effects in the treatment of viral diseases such as tick-borne encephalitis, renal syndrome hemorrhagic fever, influenza, and hepatitis B-C [38]. Srinivasan et al. synthesized a series of novel spiro-piperidinyl pyrazolones, several of which exhibited significant antiviral activity against buffalo orthopox virus (BPXV) [39]. Pyrazolines are structurally similar to pyrazolones and their derivatives also exhibit significant antiviral activity. Rizvi et al. obtained a series of novel 2-pyrazolines based on the piperidyl-thienyl ring system, using synthesis, and found that most of them have anti-HIV-1 activity [40]. Evidence has confirmed that triaryl pyrazoline impedes flavivirus infections such as dengue virus, yellow fever virus, St. Louis encephalitis virus, Western equine encephalitis virus, mouse hepatitis virus, and vesicular stomatitis virus [41]. Seeing that pyrazolines have an antiviral effect, we want to investigate whether edaravone also has an antiviral function.

We first evaluated the cell safety concentration of edaravone, because drugs have their own safe working concentrations, and if the drug concentration is too high, it can have toxic effects on the host cells. We found that the safe concentration of edaravone on GS cells was 50 µg/mL. When the concentration of edaravone was higher than 50 µg/mL, the morphology of GS cells showed pathological changes and the cell viability decreased compared with normal GS cells, indicating that the toxicity of edaravone to cells exceeded a certain concentration. However, the concentration of edaravone at 50 µg/mL had no significant toxic effects on the cell viability and morphology. After determining the safe working concentration of edaravone, we analyzed its inhibitory effect on SGIV infection. Results showed a dose-dependent inhibitory effect of edaravone against SGIV infection.

The cytoskeleton is an interconnected network of filamentous polymers and regulatory proteins composed of different types of actin, microtubules and intermediate filaments. The cytoskeleton is responsible for cell shape, whole-cell motility, and organelle motility, thus playing an important role in various cellular functions. Common cytoskeletal defects, alterations in microtubule stability, axonal transport, and actin kinetics have been demonstrated in diseased cells [42,43,44]. Research pointed out that virus infection was associated with the cytoskeleton [27]. For instance, spring viraemia of carp virus (SVCV) infection induces the collapse of the cytoskeletal fiber system and the ring structure, and filament depolymerization [44]. Given that, we speculate that if the drug has an impact on the cytoskeleton, it will promote viral infection. Thus, we investigated whether edaravone at safe working concentrations had an impact on the cytoskeleton. We observed that the GS cytoskeleton remained normal after incubation with 50 µg/mL of edaravone, indicating that edaravone at safe working concentrations had no toxic effect on the cells. The results of cell morphology were consistent with cell viability assays.

Viral infection is divided into four processes, including binding, invasion, replication, and release. The life cycle of a virus begins with the attachment of the virus to the cell membrane, followed by invasion of the host cell, intracellular replication and release of viral particles from the host cell. A good understanding of how antiviral drugs exert their antiviral effects is essential for the development of antiviral drugs [45]. It has been found that some drugs can interfere with the viral infection process and thus inhibit viral infection. For example, Xiao et al. showed that Philippine violet showed a good inhibitory effect on SGIV during the intracellular binding and replication phase of viral infection in host cells [4]. Liu et al. demonstrated that quercetin exerted antiviral effects during the SGIV binding phase to cell receptors on the cell membrane [27]. Francesc et al. found that triaryl pyrazoline worked in the dengue 1 virus replication subsystem, showing that the drug exerted a broad anti-flavivirus activity by weakly inhibiting viral translation but significantly inhibiting viral RNA synthesis [41]. Meanwhile, Brankovic et al. investigated the effect of pyrazolinones on SARS-CoV-2 entry into host cells and replication-related proteins. They found that pyrazolinones had a high affinity for M_pro_ and PL_pro_ of SARS-CoV-2 and exerted antiviral activity by hindering virus–host cell binding and virus spread [46]. In this study, we discovered that edaravone was able to interfere with binding to host cells and replication in host cells during SGIV infection; this antiviral mechanism is similar to the drug described above. Additionally, Obakachi et al. found that edaravone showed excellent binding ability and high affinity for SARS-CoV-2 spike glycoprotein (S_gp_) and host-cell human angiotensin-converting enzyme 2 protein (hACE-2), which effectively blocked SARS-CoV-2 entry into cells [47]. However, we could not find any effect of edaravone on SGIV entry into host cells.

The intact viral structure is essential for the infective ability of the virus. Liu et al. found that quercetin had a damaging effect on the particles of SGIV, thereby inhibiting the infective ability of SGIV [27]. Cheng et al. used EGCG to treat the LMBV virus and found a significant decrease in the infectivity of LMBV [48]. In the present study, we likewise evaluated the effect of edaravone on SGIV particles. It was shown that pretreatment of the virus with edaravone resulted in a reduction in viral infectivity, indirectly indicating that edaravone disrupted SGIV particles.

The innate immune response is the first line of defense of cells against viral invasion. When a virus invades, the IRF family related to interferon (IFN) can be activated and subsequently induce the expression of downstream ISG genes and transcription factors to suppress viral invasion [49]. Wang et al. found that curcumin modulates NF-κB signaling pathway-related cellular immune and inflammatory responses to exert the antiviral activity; they also found that curcumin could enhance the cellular antioxidant capacity through Nrf2/Keap1 (a typical antioxidant pathway) [50]. It was reported that edaravone could also effectively inhibit free radical-mediated tissue damage by activating Nrf2/HO-1 and inhibiting the NFκB signaling pathway [24]. In addition, SGIV infection was able to increase the production of ROS and damage GS cells [51]. Similarly, edaravone also served as a broad-spectrum antioxidant, which could scavenge ROS and RNS from aqueous and lipid environments [21]. Whether edaravone can inhibit SGIV infection through IFN pathway, the NF-κB signaling pathway, or the antioxidant pathway, needs further investigation.

## 5. Conclusions

Edaravone has a concentration-dependent inhibitory effect on SGIV infection. Moreover, edaravone can damage SGIV particles and can interfere with the SGIV infection processes of binding to host cells and replication in host cells. Our results confirm that edaravone has great potential to be a drug in the prevention and control of SGIV. 

## Figures and Tables

**Figure 1 viruses-15-02237-f001:**
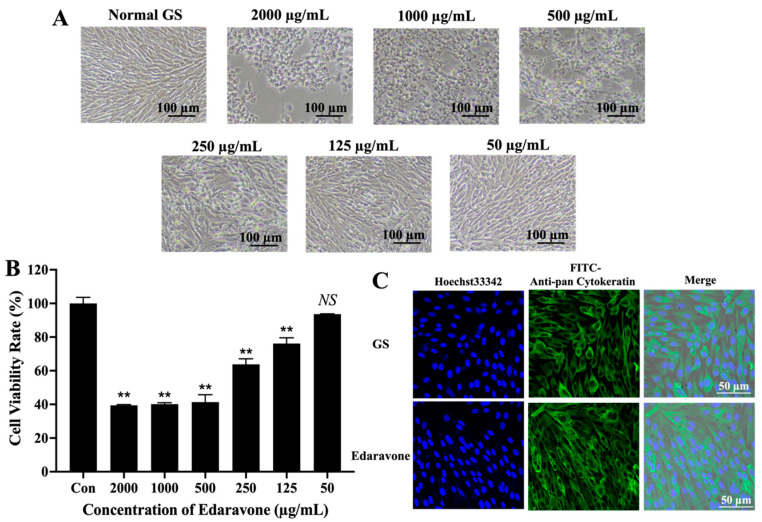
Safe cell concentration of edaravone. (**A**) It was observed by microscopy that the cell morphology changed significantly at the concentrations higher than 50 µg/mL. Scale: 100 μm. (**B**) Cell activity analysis also revealed that the concentration of 50 µg/mL had no effect on the cell activity. Data are shown as the mean SD (*n* = 3), ** indicating *p* ≤ 0.01, Ns indicating *p* > 0.05. (**C**) The cytoskeleton of the cells incubated with edaravone (50 μg/mL) remained normal compared with the control cells. Scale: 50 μm.

**Figure 2 viruses-15-02237-f002:**
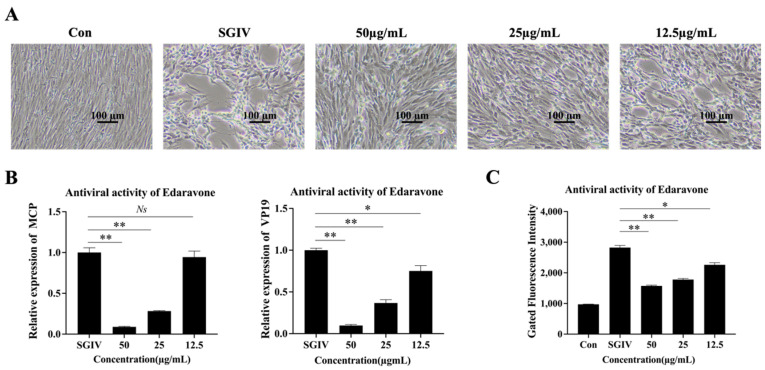
Inhibition effects of edaravone on SGIV infection. (**A**) Cell morphology was observed using light microscopy. Few typical cytopathic effects (CPEs) were observed in cells cultured with SGIV + edaravone (50, 25 µg/mL). Scale: 100 μm. (**B**) The expression of the SGIV MCP gene was significantly decreased, 11.44-fold (50 µg/mL) and 3.77-fold (25 µg/mL), and the SGIV VP19 gene was significantly decreased, 10.22-fold (50 µg/mL) and 2.72-fold (25 µg/mL) in the edaravone treated group compared to the control. (**C**) Flow cytometry assays showed a significant decrease of 3.08-fold (50 µg/mL) and 2.29-fold (25 µg/mL) in fluorescence intensity in cells treated with edaravone. Data are shown as the mean SD (*n* = 3), ** indicating *p* ≤ 0.01, * indicating *p* ≤ 0.05, and Ns indicating *p* > 0.05.

**Figure 3 viruses-15-02237-f003:**
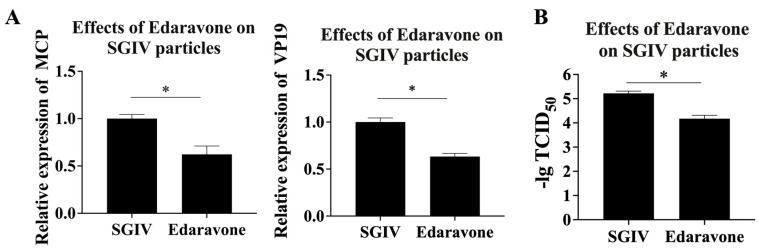
Edaravone has a destructive effect on SGIV particles. (**A**) The expression of SGIV *MCP* gene was significantly decreased, 1.6-fold, and the SGIV VP19 gene was significantly decreased, 1.57-fold, in the edaravone (50 µg/mL) treated group compared to the control. (**B**) The SGIV -lgTCID_50_ in the test group was significantly decreased, 1.25-fold. Data are shown as the mean SD (*n* = 3), with * indicating *p* ≤ 0.05.

**Figure 4 viruses-15-02237-f004:**
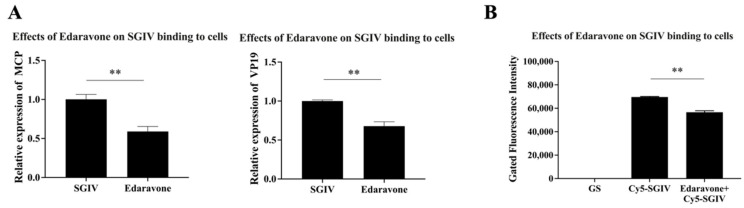
Inhibitory effect of edaravone on the binding of SGIV to host cells. (**A**) The expression of the SGIV MCP gene was significantly decreased, 1.7-fold, and the SGIV VP19 gene was significantly decreased, 1.48-fold in the edaravone (50 µg/mL) treated group compared to the control. (**B**) Flow cytometry results showed that the Cy5 fluorescence signal was significantly decreased, 1.14-fold, in cells treated with edaravone (50 µg/mL) compared with the control group. Data are shown as the mean SD (*n* = 3), with ** indicating *p* ≤ 0.01.

**Figure 5 viruses-15-02237-f005:**
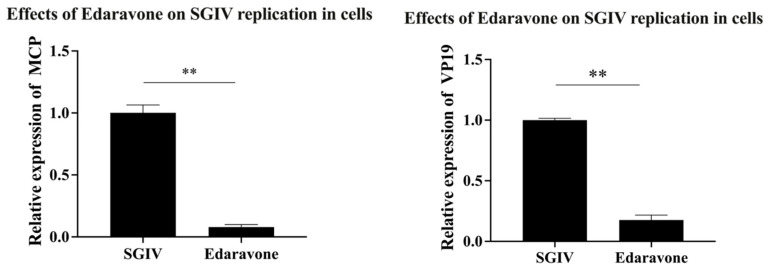
Inhibition of SGIV replication in host cells by edaravone. The expression of the SGIV MCP gene was significantly decreased, 11.53-fold, and the SGIV VP19 gene was significantly decreased, 5.71-fold, in the edaravone (50 µg/mL) treated group compared to the control. Data are shown as the mean SD (*n* = 3), with ** indicating *p* ≤ 0.01.

**Figure 6 viruses-15-02237-f006:**
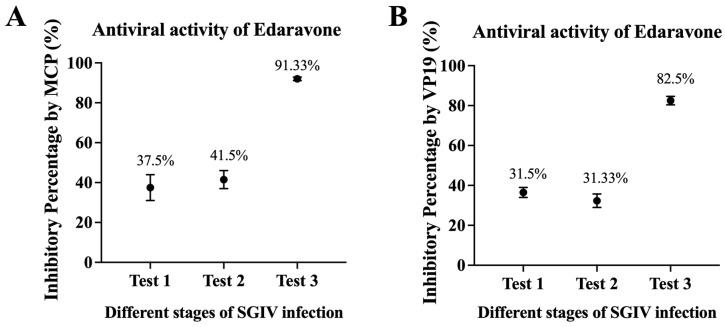
(**A**) According to the expression of viral *MCP* gene, it was shown that the inhibition rates of edaravone (50 µg/mL) on SGIV particles (Test 1), SGIV binding to host cells (Test 2), and SGIV replication (Test 3), were 37.5%, 41.5%, and 91.33%, respectively. (**B**) Similar results appeared in the expression of the SGIV *VP19* genes detection at different stages of SGIV infection.

**Table 1 viruses-15-02237-t001:** The primers used for detecting Singapore grouper iridovirus (SGIV) infection in RT-qPCR.

Primer	Sequences
q*MCP*-F	5′-GCACGCTTCTCTCACCTTCA-3′
q*MCP*-R	5′-AACGGCAACGGGAGCACTA-3′
q*VP19*-F	5′-TCCAAGGGAGAAACTGTAAG-3′
q*VP19*-R	5′-GGGGTAAGCGTGAAGACT-3′
β-actin-F	5′-TACGAGCTGCCTGACGGACA-3′
β-actin-R	5′-GGCTGTGATCTCCTTCTGCA-3′

## Data Availability

The data that support the findings of this study are available from the corresponding author upon reasonable request.

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
