# Peer review of "Antiviral Effect and Mechanism of Edaravone against Grouper Iridovirus Infection"

_viruses, 2023, doi:10.3390/v15112237_

Round 1

Reviewer 1 Report

Comments and Suggestions for Authors

Dear Editor,

The manuscript entitled “Antiviral Effect and Mechanism of Edaravone Against Grouper Iridovirus Infection” by Jihui Kuang et al. presents a study focused on evaluating whether edaravone has an antiviral effect against Singapore grouper iridovirus (SGIV), and further exploring the anti-SGIV mechanism of action for edaravone.

Τhe manuscripts’ objectives are quite interesting, the manuscript is well-written and could be accepted for publication after major revisions. My detailed comments for the authors to consider are provided below:

Major points:

1.      More concentrations should be tested to define the highest non toxic concentration. The use of 50 µg/mL as a nontoxic point and 125 µg/mL as the next point (which proves to be toxic) seems random to me. At least one more intermediate point (75-85 µg/mL) should be added to the study to be more precise on the non-toxic range which can be used.

2.      Sections 2.3 – 2.8: Please specify what concentration of edaravone was used in each case. Have you use 25 of 50 µg/mL? Its not specified either in M&M or the results section.

3.      Page 4, section 2.9: How many experiments have resulted in the presented SD? I could not find that information in any figure, and in figure 3B I see only two replicates. Please clarify.

4.      Section 3.6: I do not understand how these tests can be quantively compared, since they have different experimental parameters e.g. incubation times and drug addition points. Please explain your rational.

Minor points:

5.      Page 1, lines 40-44: appropriate reference(s) should be added.

6.      Page 2, lines 88-91: please add some more detail to facilitate readers’ understanding of the study objective, eg the study used in vitro or in vivo testing, etc.

7.      M&M section: Please add all microscopes specific types, models and providers.

8.      Page 3, lines 108-109: please add more details for the enzyme marker methodology or the appropriate reference.

9.      Page 3, line 128: what AFMP stands for? Please explain abbreviations when first appear in the text.

10.  Page 4, line 156: please add the appropriate reference or supplier for Cy5-SGIV.

11.  In figure legends please describe the figures not the results of the respective observations

12.  Page 9, line 347: please note that the SGIV disruption was proved indirectly. That result could be definitive if you had visual data, e.g. TEM image.

13.  I cannot retrieve reference 2 after performing extensive searching. Please correct it or use another reference.

Reviewer 2 Report

Comments and Suggestions for Authors

This paper is valuable in that it demonstrates the potential for the development of therapeutic agents against viral diseases in farmed fish.

page3 L97-98: Epinephelus fuscoguttatus♀ Epinephelus lanceolatus →   Epinephelus fuscoguttatus♀× Ephinephelus lanceolatus ♂

Page3 L124: major coat protein (MCP) → major capsid protein (MCP)

page9 L311: spring carp virus (SVCV) → spring viraemia of carp virus (SVCV)

page9 L345: shouwed → showed

Round 2

Reviewer 1 Report

Comments and Suggestions for Authors

I thank the reviewers for incorporating my comments in their manuscript